



# Lessons from the 2018-2019 European droughts: A collective need for unifying drought risk management

Veit Blauhut[1], Michael Stoelzle[1], Lauri Ahopelto[2,3], Manuela I. Brunner[1], Claudia Teutschbein[4], Doris E Wendt[5], Vytautas Akstinas[6], Sigrid J. Bakke[7], Lucy J. Barker[8], Lenka Bartošová[9], Agrita Briede[10], Carmelo Cammalleri[11], Lucia De Stefano[12], Miriam Fendeková[13], David C. Finger[14,15], Marijke Huysmans[16], Mirjana Ivanov[17], Jaak Jaagus[18], Jiří Jakubínský[9], Ksenija Cindrić Kalin[19], Svitlana Krakovska[20], Gregor Laaha[21], Monika Lakatos[22], Kiril Manevski[23], Mathias Neumann Andersen[23], Nina Nikolova[24], Marzena Osuch[25], Pieter van Oel[26], Kalina Radeva[24], Renata J. Romanowicz[25], Elena Toth[27], Mirek Trnka[9], Marko Urošev[28], Julia Urquijo Reguera[29], Eric Sauquet[30], Silvana Stevkova[31], Lena M Tallaksen[7], Iryna Trofimova[20], Michelle T. H. van Vliet[32], Jean-Philippe Vidal[30], Niko Wanders[26], Micha Werner[33], Patrick Willems[34], Nenad Živković[35]

[1] Environmental Hydrological Systems, Faculty of Environment and Natural Resources, University of Freiburg, Germany
[2] Water and Development Research Group, School of Engineering, Aalto University, Finland
[3] Freshwater Centre, Finnish Environment Institute, Finland
[4] Department of Earth Sciences, Program for Air, Water and Landscape Sciences; Hydrology, Uppsala University, Sweden
[5] School of Geography, Earth and Environmental Sciences, University of Birmingham, UK
[6] Laboratory of Hydrology, Lithuanian Energy Institute, Lithuania
[7] Department of Geosciences, University of Oslo, Oslo, Norway
[8] UK Centre for Ecology & Hydrology, Wallingford, UK
[9] Global Change Research Institute CAS, Brno, Czech Republic
[10] Faculty of Geography and Earth Sciences, University of Latvia, Latvia
[11] European Commission, Joint Research Centre, Ispra, Italy
[12] Facultad de Ciencias Geológicas, Universidad Complutense de Madrid, Spain; and Water Observatory, Botín Foundation, Madrid, Spain; luciads@geo.ucm.es
[13] Miriam Fendeková, Univerzita Komenského v Bratislave, Slovak Republic
[14] School of engineering, Reykjavik University, Reykjavik, Iceland, fingerd@gmx.net
[15] Energieinstitut an der Johannes Kepler Universität, Linz, Austria
[16] Department of Hydrology and Hydraulic Engineering, Vrije Universiteit Brussel, Brussel, Belgium
[17] Institute of Hydrometeorology and Seismology, Montenegro
[18] Department of Geography, Institute of Ecology and Earth Sciences, University of Tartu, Estonia
[19] Croatian Meteorological and Hydrological Service, Zagreb, Croatia
[20] Laboratory of Applied Climatology, Ukrainian Hydrometeorological Institute, Kyiv, Ukraine
[21] Institute of Statistics, University of Natural Resources and Live Sciences, Vienna, Austria
[22] Hungarian Meteorological Service, Budapest, Hungary
[23] Department of Agroecology, Aarhus University, 8830 Tjele, Denmark
[24] Faculty of Geology and Geography, Sofia University "St. Kliment Ohridski", Bulgaria





[25] Institute of Geophysics, Polish Academy of Sciences, Warsaw, Poland
[26] Water Resources Management Group, Wageningen University, the Netherlands
[27] Dept. of Civil, Chemical, Environmental and Materials Engineering, University of Bologna, Bologna, Italy
[28] Geographical Institute "Jovan Cvijić", Serbian Academy of Sciences and Arts, Serbia
[29] Department of Agroforestry Engineering, Escuela Técnica Superior de Ingeniería Agronómica y de Biosistemas, Universidad Politécnica de Madrid, 28040 Madrid, Spain
[30] INRAE, RiverLy, Villeurbanne, France
[31] Department of Meteorology, National Hydrometeorological Service, 1000 Skopje, North Macedonia
[32] Department of Physical Geography, Utrecht University, The Netherlands
[33] IHE Delft Institute for Water Education, Water Resources & Ecosystems Department, Delft, the Netherlands
[34] Hydraulics and Geotechnics Section, Department of Civil Engineering, KU Leuven, Belgium
[35] Faculty of Geography, University of Belgrade, Serbia

*Correspondence to*: Veit Blauhut (veit.blauhut@hydrology.uni-freiburg.de)

**Abstract.** Drought events and their impacts vary spatially and temporally due to diverse pedo-climatic and hydrologic conditions, as well as variations in exposure and vulnerability, such as demographics and response actions. While hazardous
severity and frequency of past drought events have been studied in detail, little is known about the effect of drought management strategies on the actual impacts, and how the hazard is perceived by relevant stakeholders for inducing action. In a continental study, we characterised and assessed the impacts and the perceptions of two recent drought events (2018 and 2019) in Europe and examined the relationship between management strategies and drought perception, hazard and impacts. The study was based on a pan-European survey involving national representatives from 28 countries and relevant stakeholders
responding to a standard questionnaire. The survey focused on collecting information on stakeholders' perceptions of drought, impacts on water resources and beyond, water availability and current drought management strategies at national and regional scales. The survey results were compared with the actual drought hazard information registered by the European Drought Observatory (EDO) for 2018 and 2019. The results highlighted high diversity in drought perceptions across different countries and in values of implemented drought management strategies to alleviate impacts by increasing national and sub-national
awareness and resilience. The study concludes with an urgent need to further reduce drought impacts by constructing and implementing a European macro-level drought governance approach, such as a directive, which would strengthen national drought management and lessen harm to human and natural potentials.

# 1 Introduction

## 1.1 Drought impacts in Europe

During recent decades, Europe has been affected by a number of severe, large-scale drought events, e.g. in 2003, 2007, 2011, 2012, 2015, 2017, 2018, 2019 and 2020 (Baruth et al. 2020, Boergens 2020, 2017; Cindrić et al. 2015, Garcia-Herrera et al. 2019; Hänsel et al. 2019; Ionita et al, 2017; Laaha et al., 2017). Each of these droughts was unique in terms of severity, spatio-temporal extent and associated direct and indirect impacts to human and natural resources (Stahl et al. 2015). Cammalleri et



al. (2020) estimated drought-related losses in the European Union (EU) to about 9 billion Euros annually. The largest share of these losses is typically seen in agriculture, energy and public water supply sectors (Cammalleri et al. 2020), triggered mainly by agricultural (soil moisture deficit) and hydrological drought (deficit in river flow and groundwater; Van Lanen et al. 2015). These sectoral losses likely represent only part of the actual drought impacts, simply because indirect, intangible or subtle impacts are more difficult to identify and quantify, such as adverse effects on ecosystem services and human health (UNDRR, 2021). According to the European Drought Impact report Inventory (EDII; Stahl et al. 2016), further impacts on aquaculture, ecosystems, humans and public safety, as well as conflicts between sectoral water users, have been reported. Herein, the occurrence and the composition of drought impacts are assumed to greatly vary with regional and national exposure, perception and vulnerability to droughts (e.g. Stahl et al., 2016).

## 1.2 Drought Management in Europe

A key element to mitigate drought impact is to respond promptly, i.e., implement drought management planning strategies and associated action plans (UNDRR 2019). However, a directive for drought risk management does not exist on near-continental scale such as in the EU (Hervás-Gámez et al. 2019), despite the identified potential for reducing emergency management costs through proactive management (Cammalleri et al 2020; Howarth et al. 2018). So far, "droughts have only been succinctly dealt with in the Water Framework Directive with no compulsory actions" (Hervás-Gámez et al. 2019). However, recommendations are not adopted in all relevant/major river basin districts (EC 2019). The "European Commission's Communication on water scarcity and drought" and the 'Blueprint to Safeguard Europe's Water Resources' (EC 2012) directly tackle drought and address current flaws and policy gaps. These documents have received a mixed response, ranging from "the Communication is still weak and lacks teeth in the policy landscape" (Stein et al. 2016) to "it is hoped to lead to an EU water policy development in a long term" (Hervás-Gámez et al. 2019). However, some countries have historically been and are more prone to drought compared to others due to their pedo-climatic settings, and although drought risk management does exist in these countries through national legislation, it mostly happens indirectly via policy-making regarding environmental protection, soil management, or water and climate adaptation (e.g. Caillet et al. 2019, Hanger-Kopp and Palka 2020). Moreover, a number of technical guidelines exist to support the development and the implementation of national drought resilience, adaptation and management plans (e.g. UNCCD 2019). In fact, different national legislations not being internationally coordinated can create conflicts, i.e., water scarcity in one region/country at the cost of another, such as the case of the Blue Nile between Egypt and Ethiopia (El Bastawesy et al., 2015) or the Danube between Hungary and Slovakia (Vuković et al., 2014). Therefore, a coordinated approach is required. Trnka et al. (2018) list improving the understanding of triggers causing paradigm shifts from response-based to proactive drought management and policies as a priority research question.

## 1.3 The 2018 & 2019 European droughts

For several successive years, large parts of Europe were affected by severe and widespread summer drought, which highlighted the vulnerability of its socio-economic and environmental systems. The 2018 event was special because of both rainfall deficit



and high temperatures in many European countries (Rosner et. al 2019), with record-breaking high temperatures in several regions (Bakke et al., 2020), and reached otherwise cool and humid northern regions. This compound hot-dry event led to major impacts in north-central and north-eastern Europe, particularly affecting agriculture, livestock farming and forestry (Bakke et al. 2020, Beillouin et al. 2020, Rosner et al. 2019, Salmoral et al. 2020, Schuldt et al. 2020, Thompson et al. 2020)

as reported for Sweden, Finland, Estonia, Lithuania, Latvia, Denmark, the Netherlands, Belgium, Germany, the United Kingdom, and eastern France (Moravec et al. 2021, Turner et al. 2021). The propagation of the meteorological drought resulted in low reservoir levels and river discharge, which impaired public water supply, leading to partial shut downs of nuclear power plants and triggering massive fish deaths in upstream watersheds (e.g. de Brito 2021). In contrast to central and northern Europe, the western Mediterranean countries experienced above-average wet conditions in 2018 after having experienced a

very severe drought on the Iberian Peninsula in 2016–2017 and in Italy in 2017 (Garcia-Herrera et al. 2019; Rita et al. 2019), while the eastern Mediterranean experienced below average dry conditions (DriDanube-Watch 2018). Opposite to 2018, the 2019 drought was centred on eastern Germany, Czech Republic and Poland before spreading westward (Boergens et al. 2020). The most affected regions were still suffering from large water balance deficits from the 2018 drought (Boergens et al. 2020) at the start of 2019. Hari et al. (2020) declared the period 2018–2019 in central Europe a two-year drought event unprecedented

in severity in the last 250 years, whereas Büntgen et al. (2021) show an accumulation of drought signals in central Europe over five summers i.e. 2014-2018.

### 1.4. Drought risk and perception

The hydro-climatic aspects of past drought events have been studied in detail (e.g. Barker et al. 2019; Hisdal and Tallaksen 2003; Dai 2013; Cheval et al. 2014; Jaagus et al. 2021; Laaha et al. 2017; Radeva et al. 2018; Spinoni et. al. 2015, 2018),

whereas knowledge of the relationship between drought management, perception and impacts remains limited (Blauhut 2020, Hagenlocher et al. 2019; Kreibich et al. 2019). Understanding how different stakeholders perceive a specific drought event and its potential impacts can contribute to defining and successfully implementing drought mitigation measures adapted to a site-specific context (Alduce et al. 2017). Only a few studies have analysed relationships between drought perceptions and impacts. For instance, Teutschbein et al. (2019) assessed the link between perceived drought severity, impacts, preparedness

and management and measured hydrological drought impacts for two consecutive drought events (2017 and 2018) in Sweden. Although the authors did not find a significant relationship between the perceived level of drought impacts and the presence of a drought action plan, there was evidence that regions with a drought action plan applied significantly more measures in their drought response. Furthermore, the perceived drought severity did not match the observed severity of meteorological and hydrological droughts in Sweden: decision makers consistently overestimated the severity of mild drought events, while

underestimating more extreme drought conditions. In contrast, Blauhut et al. (2016) identified "drought awareness" and "drought management plans" as vulnerability factors driving drought risk for certain impact categories, such as agriculture and livestock farming, public water supply or freshwater ecosystems. The analysis of Blauhut et al. (2015b) suggested that while national and international water management policies and guidelines may have decreased vulnerability, they may also have



increased awareness and recognition of environmental impacts, leading to an increased number of reported drought impacts.
Hence, previous statements on the relationship between the existence of drought risk management plans and drought impacts
cannot be generalised.

### 1.5. Study aim

The aim of this paper is to assess how monitored drought-hazard severity relates to drought perception and drought
management strategies. We hypothesise that perceived drought impacts are not necessarily related to the severity of the drought
hazard, but are strongly influenced by national awareness and drought management strategies. To verify this hypothesis, we
investigated how the droughts of 2018 and 2019 in 28 European countries were related to a) the drought hazard as monitored
by the European Drought Observatory (EDO; https://edo.jrc.ec.europa.eu), b) drought management actions taken in the
different countries, c) drought perception by water managers and agencies and d) drought awareness. National drought
perceptions, management and impacts were studied using a pan-European survey. On the basis of this survey, we discuss the
potential benefits of a European drought directive, similar to the Floods Directive (2007/60/EC) with respect to reducing
drought vulnerability and impacts by macro-level governance.

### 2. Data

In order to evaluate the hypothesis, two different types of spatial data were collected and compared: i) drought information as
monitored by the EDO (https://edo.jrc.ec.europa.eu) and ii) information on drought impacts, perception and state of drought
management plans collected through a pan-European survey targeting water managers and water agencies.
The hydro-climatic situations in 2018 and 2019 were described using a set of drought indices compiled by EDO for a variety
of drought types including meteorological drought (Standardised Precipitation Index (SPI) for 1, 3, 6, 9 and 12 month
accumulation periods), soil moisture drought (soil moisture anomaly; SM), hydrological drought (Low Flow Index; LFI,
representing the discharge anomaly with respect to a daily threshold), and vegetation drought (anomaly of Fraction of Absorbed
Photosynthetically Active Radiation; fAPAR). The SPI is given at a monthly resolution, whereas the other indices are presented
in 10-day non-overlapping intervals. Detailed information on the drought indices can be found in the EDO indicator factsheets
(https://edo.jrc.ec.europa.eu/edov2/php/index.php?id=1101). To increase comparability of the four indices, the EDO data was
further classified into categorical drought classes according to the thresholds listed in Table 1. Furthermore, the fAPAR was
restricted to the warm season in Europe from April to August and was not monitored for Iceland.
In order to assess the country specific perception of drought, management and impacts with focus on the 2018 and 2019, a
pan-European survey was designed by the International Association of Hydrological Science (IAHS) - Panta-Rhei "Drought
in the Anthropocene" working group. National representatives of each country were selected and assigned responsibility to
translate, distribute and evaluate the survey and all associated communication and feedback. The survey targeted
representatives of water management organizations and water agencies. Survey respondents were selected by the national
representatives aiming to provide a balanced view of national opinions and drought management practices (or actions), as well





as local and regional knowledge within each country. The content of the survey was adapted from Teutschbein et al. (2019), who studied 290 Swedish municipalities to evaluate the relationship between perceived drought severity, impact, preparedness and management, aiming to compare stakeholder perception with hydrological drought indices. The perception of heat was not investigated.

• The 26 questions of the survey covered the following themes:
       • Respondent background and water resource(s) used/managed,
       • General perception of drought and associated risks,
       • Drought risk-related concepts and the drought management applied, and
       • Perception of the 2018 and the 2019 drought events and their impacts

The survey questions can be found in Table S1 in the supplements. The paper and the figures displayed in the main body present a synthesis and insights from the pan-European comparison of the responses. More detailed aspects of the individual country responses are shown in Figs. S1-7 of the supplements.

### 3. Results and Discussion

### 3.1. The drought events of 2018 and 2019 – hydro-meteorological results

The drought indices of the pan-European droughts in 2018 and 2019 are presented in Fig. 1, showing the specific hydro-meteorological conditions of a specific time in the year or the month with maximum proportion of area within each country affected by the hazard, as defined by severe or extreme category. The results on monthly and national resolution are shown in Fig. S1 in the supplements. Overall, the 2018 meteorological drought (as defined by SPI-3, SPI-6, SPI-9 and SPI-12) affected mainly Central and Northern Europe. The Benelux countries, Germany, Denmark, Sweden and Finland showed an especially

high spatial coverage of severe or extreme drought hazard. In early spring, rainfall deficits started in the North, i.e., Norway, Sweden and Finland, Lithuania and Latvia (Fig. S1) and accumulated over Central and Northern Europe, peaking in the summer with high shares of extreme drought hazard at short accumulation periods (SPI-1). Strong soil moisture deficits in the summer were detected in regions affected by strong precipitation deficits over multiple months (SPI-3, SPI-6; Fig. S1). At the European scale, soil moisture deficits were especially high in Northern Europe from June to August and the area under severe

drought in central Europe peaked in October and November (Fig. 1).

The hydrological drought of 2018 followed a similar spatial pattern as the meteorological drought, with severe hazard levels in the Benelux countries, Germany, Czech Republic, Norway and Sweden. The maximum spatial coverage of severe or extreme low flows in Northern Europe occurred in June and July, in particular for countries where rainfall deficits continued and more intense deficits developed (SPI-9, SPI-12; Fig. S1). The maximum coverage of the 2018 hydrological drought in Central

Europe occurred in October and November (Fig. 1).

Vegetation drought indicated by fAPAR was the most severe in Denmark and was contrasting with the drought signals of the other indices. For example, large parts of Belarus and France were under severe or extreme drought, while only small parts of



Sweden and Finland were affected. In countries where precipitation deficits continued to accumulate over the 2018/2019 winter period, water deficits resulted in country-specific low flow conditions (Fig. 1). The multi-year drought 2018-2019 particularly

affected Belgium, Belarus, the Czech Republic, Germany, Finland, Latvia, Luxemburg, Lithuania, the Netherlands, Poland, Sweden, Switzerland and the United Kingdom. Furthermore, Iceland also experienced an exceptionally long dry period in 2019. However, the effects were not as intense as in mainland Europe, mainly due to the numerous icecaps that provided ice melt, an extensive snow cover during winter (Helmert et al. 2018), a subpolar climate, and warm and humid ocean winds, that could generate local rain events (Finger, 2018).

The 2019 drought was overall less severe compared to 2018, except for Iceland, who experienced an unusually long dry period. The centre of the meteorological drought moved eastwards with large areas under severe or extreme drought in Lithuania, Belarus and Ukraine. Despite the lower intensity of the 2019 meteorological drought, soil moisture deficits remained high in Central Europe, especially Poland, the Baltic and the Benelux countries. In Central Europe, soil moisture drought peaked in early February and March compared to a delayed peak in April in Poland and the Baltics and an even later peak in Ukraine

and Moldova towards the end of 2019. The low flow situation in Central Europe and Scandinavia partly recovered, although severe hazard levels were still detected from July to September. The eastern European countries showed an overall increase in low flow severity peaking earlier in the year (April and May). In addition, fAPAR was less severe in 2019 for most months and most of Europe, while South-Eastern Europe and the Balkans showed increased hazard severity.

In southern and south-eastern Europe, the hydro-climatic conditions of 2018 and 2019 differed from the rest of Europe. In

2018, Spain, Portugal and Italy had recovered from drought conditions, but deficits again developed in early 2019. In south-eastern Europe, winter 2018/2019 precipitation deficits were detected across much of the Balkan Peninsula, i.e., Croatia, Albania, Slovenia, North Macedonia, Montenegro and Hungary, as well as in Slovakia. Serbia was already affected by the summer drought in 2018, which persisted throughout the winter. In Ukraine, Moldova and Romania, the 2018 event was moderate in the second half of the year and further rainfall deficits accumulated during winter, which led to rising soil moisture

deficits from summer 2019 to the end of 2020.

### 3.2. The drought events of 2018 and 2019 - perception and management

The online survey yielded contributions by 712 respondents from 28 European countries (Fig. 2a) with the number of responses varying by country, i.e., from a single expert (Romania) to over 100 replies (Sweden). The majority of the respondents were employees at governmental institutions (74%) at different administrative levels, with expertise related to water management,

environment, meteorology and agriculture. Furthermore, private and public companies (operators of public water supply systems, hydropower plants; 13%), scientific institutions (4%) and other non-governmental organisations with a focus on environment and ecology (3%) also contributed to the survey.

The importance of the water resources as perceived by the participants (under normal conditions) ranked differently across the continent (Figs. 2b and c). If the nationally averaged importance of water resources were ranked equally (e.g. groundwater and

individual well both ranked as second most important), their importance as rank #2 and #3 were also evaluated. Overall, the



majority of the respondents selected groundwater as the most important resource (~35% of all participants), followed by surface water from rivers (22%), reservoirs (13%), individual wells (11%) and artificial groundwater recharge (11%). Further "other" water resources such as rainfall collectors, ponds or water transfer systems were listed a few times (<1%). Specific spatial patterns of water resource importance were not apparent, although individual wells appeared to be more important in

eastern Europe and artificial groundwater recharge was highlighted in Italy, Hungary and Bulgaria. For Spain, this question was adapted to national specificities and resulted in #1 regulated surface water and #2 groundwater. A more detailed national breakdown can be found in Fig.2.

The use of a drought definition to categorise drought hazard varied markedly across Europe (Fig. 3). About 40% of all participants did not have an operational drought definition in their municipality or company, and a further 15% did not know

whether there was one. In contrast, for Czech Republic, Spain, Italy and France all participants had an operational drought reporting system. With regard to the participants' affiliation, about 60% of those working for governmental authorities did not have - or were not aware of - an operational drought definition, in contrast to private companies where around 30% were unaware of a drought definition. Overall, about 20% defined drought by a single drought type index (such as meteorological drought), 15% used two and 10% used three different drought indices. The majority of participants used meteorological and

hydrological indices (30% each) and about 15% relied on soil moisture and vegetation conditions. Furthermore, drought-impact information such as vegetation activity (e.g. NDVI), crop yields or forest fire indices, but also media reports were used. In Spain, the 'Special Drought Management Plans' define two types of drought-related events: prolonged drought (meteorological) quantified by precipitation deficit over different time periods, and conjectural water shortage identified through the assessment of available water resources. This question was not asked in Sweden and Poland, and in Latvia, drought

definitions were not operationalised.

Following the drought definition question, respondents were asked whether an established governmental drought declaration system existed or the declaration of drought situations was based on case-specific decisions (Fig. S2). An operational declaration scheme is defined here as an official government implemented method of defining a drought situation, often including drought severity thresholds and pre-defined measures. Operational drought declaration schemes (at country or county

level) were scarce across the continent, though these were found to be present in Spain, France, the Netherlands and the Czech Republic. In Czech Republic, drought declaration is based on the open national drought monitoring platform Intersucho (Trnka et al. 2020). The same platform is shared also by the Slovak Hydrometeorological Institute (Labudová et al. 2018).

In Spain, governmental drought declaration schemes are included in the Special Drought Management Plans approved at river basin level, whereas each one has adapted it to their context and characteristics. Outside of Spain, individual decisions on

drought declarations are more commonly present in regions with a lack of fixed drought declaration schemes. In some countries, drought situations are declared by the "Emergency situations commission meetings", for example in Lithuania, the Netherlands and the UK, where a national water management centre and drought committee advise the government. In Latvia, Estonia, Austria, Bulgaria and Denmark, more than half of the respondents did not know of the existence of any governmental drought declaration scheme or were not sure that one existed.



In over 25 countries, the majority of participants (>50%) responded positively to the question of whether future climate change may affect water resources. (Fig. 4 a). The majority of respondents expected the occurrence of droughts to "increase" or "strongly increase" in the (near-) future (Fig. 4 b). However, no relationship could be established between the expected future changes in drought hazard and the degree to which climate change is considered in policies. In addition, the responses about the expectation of "the need for more regulation of water distribution to fewer consumers due to shortages in the future" was

neither linked to the expectation of future drought occurrence nor climate change consideration (Fig. 4 c). For example, in the UK, around 15% of respondents agreed that more regulation will be needed with a majority expecting an increase in drought occurrence, whereas in North Macedonia, about 85% agreed on a need for increased future regulation, having a similar share in future drought occurrence. Nevertheless, it appears that participants of northern European countries perceive less need for more regulation compared to the rest of Europe. As mentioned by the participants, future regulation is expected to take the

form of an EU drought directive, ranking priorities, re-allocating water permits, technological enhancements to save water, water pricing and general water usage restrictions.

Few participants indicated that their countries had drought management action plans (~10%), although emergency action plans were more common (~25%), and both plans existed more in western compared to eastern Europe (Fig. 5 a). The UK, the Netherlands, Belgium, Spain, Switzerland, Italy and Montenegro were comparatively well prepared in this regard (>75% of

the participants had an emergency action or management plan). Participants from Mediterranean countries such as Spain and Italy, but also Czech Republic, had a high share of drought risk management plans, with more than 150 participants indicating the intention to introduce new (or update existing) drought management plans. As indicated by the participants, such management strategies range from the "simple increase" of water storage reservoirs or adapting farming practises, to the development of legally binding drought risk management strategies.

To better understand the reasons for an absence of drought management plans, participants were asked for a possible explanation and answers were provided by the national experts as either pre-defined or free text options. At the country scale, "insufficient resources" and the perception that "drought is not seen as a risk" were the most often answers (Fig. 5b). For northern Europe and Austria, drought not seen as a risk was highlighted most often, whereas for the eastern European countries, the "lack of legal obligation- no European drought directive" and "Waiting for governmental advice" were selected by about

15% of all participants. Further, a "Lack of knowledge" (on drought risk) was more prevalent in western Europe, whereas a "Lack of resources" (finance and capacity) were prominent in central and south-eastern Europe. Political issues (waiting for governmental advice, lack of forcing – no EU drought directive, and political ignorance) were especially present in central and eastern Europe, but were less prominent in northern Europe.

With regard to communication and interaction during drought events, participants were asked whom they collaborate with to

manage droughts (Fig. S3). On average, more than half of the participants collaborated with "other authorities" (e.g., county administrative boards or water authorities). About 45% interacted with "other departments or companies within the municipality" and about 20% with land owners and independent experts (such as universities). About 20% did not know about any collaboration and 5% of participants stated no existing collaboration.





### 4.3. Survey-based perception and management of the 2018 and 2019 droughts in Europe

The perception of the 2018 and 2019 drought events by the survey respondents showed country- and event-specific differences (Fig. 6). The participants could rank the hydroclimatic situation from extremely dry to wet; 2018 was mostly perceived as "drier than normal" and central and northern European countries in particular perceived "very dry" conditions (Fig. 6 a), with high proportions (25%) of "extremely dry" conditions in Czech Republic, Germany, Lithuania, North Macedonia and Norway (Fig. S4). The south-western European countries also perceived 2018 as "drier than normal". In Iceland, 2018 was perceived

as a wet year, in contrast to the dry conditions of 2019 which were perceived to be "one of the worst droughts" on record. For the rest of Europe, the 2019 drought was perceived as less severe than 2018 with the exception of France and Ukraine with high variations between countries (Fig. 6 b). The centre of the 2018 drought event shifted by the end of the year/ beginning of 2019 from Central and northern Europe towards the East. Wetter conditions in northern Europe translated into perceptions of no or less severe drought in Scandinavia and the Baltics, respectively. The hydro-climatic situation in 2019 was still perceived

as "very dry" (50%) in France, Belgium, Germany, Slovakia and Ukraine.

Drought management preparation in 2018 showed an east-to-west gradient, i.e., eastern, northern and central European countries felt overall more "prepared" while countries in western Europe perceived they were "not well" prepared (Fig. S5). The management of the 2018 drought event was generally perceived as being worse compared to 2019, except in some central and northern European countries. Most respondents thought that they were better prepared in 2019 due to the previous event

that likely contributed to an earlier activation of emergency plans, if any. However, the perception of drought impacts only shows minor differences between the two drought events (Fig. 6 c and d), with the exception of northern Europe. The Mediterranean and the Balkan countries perceived drought impacts as not severe or without impacts (Croatia) in both years, with a tendency towards a higher severity in 2019 for black sea countries. In Central Europe, participants perceived that they were severely affected in 2018 and this perception extended towards eastern Europe in 2019. Scandinavia and the Baltics were

only slightly affected in 2018 with a lower perceived severity in 2019. For the majority of respondents, the drought of 2018 played a crucial role in the perceived impacts of the 2019 drought event (Fig. S6). Most respondents perceived particularly negative consequences for agriculture, livestock farming, forestry and public water supply in 2018 and 2019 compared to relatively minor consequences regarding air pollution and conflicts (Fig. S7). At a first glance, the perception for these sectors differs only slightly between the events. However, soil moisture impacts, such as agricultural losses, impacts on freshwater

aquaculture and fisheries or forest fires were less reported in 2019. Observing from 2018 to 2019, Denmark, Norway and the UK show stronger reductions in perceived impacts from 2018 to the 2019 drought event. Slightly more impacts were reported in 2019 for livestock farming in Ukraine, for forestry and terrestrial ecosystems in Belgium and Ukraine, for air quality in Bulgaria, Slovakia, North Macedonia and Ukraine, and for water quality in Austria, Czech Republic and North Macedonia. In contrast, Iceland was only affected in 2019 with strong effects on agriculture and water quality.



## 4. Discussion

This is the first study that quantifies drought perception by relevant stakeholders at continental scale based on participatory survey. The survey analysis shows high diversity in perceived drought impacts (Fig. S7), which reflects Europe's pedo-climatic and socio-economic heterogeneity as is also shown by Stahl et al. (2016). The monitored and perceived drought hazard differed in some places as a result of the different drivers of drought impacts: hazard, exposure and vulnerability (IPCC 2014, UNDRR 2019). The diversity of impacted categories has been reported previously for similar drought events, e.g., in 1975, 1976 or 2003 (Stahl et al. 2016). Our findings corroborate those of Stahl et al. (2016) with different countries across Europe being affected by the hazard very differently. Large-scale weather patterns and differences in land surface properties play a crucial role in explaining this heterogeneity. For instance, Atlantic meridional dipole circulation anomalies have been found to be associated with Northern European droughts as represented by the SPI-6 and SPEI-6 (Standardised Precipitation-Evapotranspiration Index) indices (Kingston et al. 2015). The Scandinavian teleconnection pattern, which was unusually high in May and July 2018, resembles the large-scale atmospheric circulation pattern most associated with summer low flow in south- and eastern Scandinavia (Bakke et al. 2020). It should be also noted that the frequency of drought-related circulation patterns has been changing since the end of the 19th century with increasing frequencies over Central Europe (e.g. Lhotka et al., 2020; Trnka et al. 2009). The unique conditions of Iceland, where major drought events cannot be compared to the rest of Europe, was also shown by Spinoni et al. (2015), most likely attributed to its location influenced by warm humid winds and enhanced by the Gulf stream clashing with the cold Arctic winds from the North that generate frequent precipitation events (de Niet, 2020). Nevertheless, severe land degradation in Iceland has decreased the water holding capacity, making the land susceptible to hydrological droughts (Finger et al. 2016, Keesstra et al. 2018). Furthermore, Spinoni et al. (2019) showed that major drought events as indicated by SPI and SPEI in central and northern Europe, north-eastern Europe, and southern Europe do not occur simultaneously, which was also evident in our results focusing on 2018 and 2019. The multiyear drought character of 2018 and 2019 became evident when focusing on the monitored hydrological drought conditions in Belgium, Switzerland, Czech Republic, Germany, Finland, Latvia, Luxemburg, Lithuania, Poland and Sweden.

In general, the hazard severity perceived by the surveyed stakeholders corresponded well with the hazard severity monitored by the EDO, though with some exceptions. For example, in 2018, large areas of Sweden and Finland were affected by severe (or extreme) hazard conditions according to the EDO, but the hydro-climatological situation was perceived as "dry". In 2019, EDO reported a severe (or extreme) meteorological drought in Iceland and a severe (or extreme) soil moisture drought in the Baltic, but in both countries, the hazard severity was perceived as less severe and indexed as "drier than normal" to "dry". In contrast, Norway's participants perceived very dry conditions in 2018, but the proportion of monitored severe (or extreme) hazard conditions were low. These discrepancies could be attributed to the low awareness of the stakeholders for the drought conditions at a larger scale or the impact across different sectors. Alternatively, standardised drought indices not effectively representing drought conditions everywhere may also contribute to these discrepancies. In higher latitude countries, a strong negative rainfall anomaly does not necessarily imply a deficit in water availability for e.g. plant water uptake or public water





supply as storages are usually replenished after the snow melt period. (Cammalleri et al. 2016). As such, meteorological drought indices may not be appropriate to predict impacts and consequences for management of hydrological or agricultural

(soil moisture) droughts. The wide range of drought definitions and associated high number of drought indices - combined with a widespread lack of operational declaration schemes - highlight the many obstacles when dealing with the complex inter- and transdisciplinary nature of drought impacts. A unique definition of drought that is valid across all regions and sectors is not possible in practice (Lloyd-Hughes 2014), especially if sectors, such as agriculture and water supply, are based on different laws and managed by different authorities. An effective implementation of macro-regional drought risk management requires

a more holistic interdisciplinary view.

Our pan-European survey reflects the opinions of water professionals belonging to mostly the public sector and publicly owned companies. The perspectives of other citizens, local stakeholders, private companies and NGOs were less well represented. Accordingly, sectoral and regional perceptions of drought risk might differ. For example, a hydropower production survey in

southern Germany showed that legislative drought risk regulation is not appreciated by reservoir operators, who would nevertheless support the development of drought risk management coupled with integrated river basin management (Siebert et al. 2021).

The preferential use of meteorological and hydrological indices to define drought by the participants was found similar to the findings of Bachmair et al. (2016). The absence of dedicated drought risk management strategies in many European countries

is evident (Fig. 5 a) due to diverse and in some cases, contradictory reasons (Fig. 5 b). The country representatives were asked some broad questions on the state of national drought management and the potential for a European drought directive; the responses revealed an unsatisfactory state of national drought risk management in Europe Fig. 7).

The existence of drought risk management plans or strategies tended to be higher in countries with more common water scarcity issues and more frequent drought events, such as those in the Mediterranean region (Tramblay et al. 2020). Moreover,

only Spain's 'Special Drought Management Plans'(updated in 2018; Hervás-Gámez, C., & Delgado-Ramos, F., 2019) was assessed as comprehensive and sufficient by national representatives. In addition, recent drought events may have forced governments to foster drought research and policy implementation, suggesting that a 'memory of recent disasters' improves disaster management and potentially mitigates drought impacts (DiBaldassarre et al. 2013; Kreibich, et al. 2017). Urquijo et al. (2016) stated that drought management is a combination of the history of water management and the frequency of drought,

which is supported by our results from the Mediterranean countries and the Netherlands. Furthermore, case-specific effects of drought may also drive the need for risk management. In the Netherlands, for example, hydrological drought can increase salt water intrusion, increase land subsidence and structural instability of dikes. The resulting damages of these hydrological drought impacts decrease water security in the long-term, especially with regard to compound events. The engagement of non-governmental scientific groups also fosters drought risk management and particularly public and government awareness (e.g.

Czech Republic).



The diversity of drought management approaches reflects the diversity of Europe's hydro-climatic conditions and governance contexts. However, droughts do not respect national borders and Europe has several shared river basins. In addition, climate change is estimated to increase drought severity and frequency globally and in Europe (UNDRR, 2021; Spinoni et al., 2018). A pan-European drought management approach would support national and cross-boundary drought preparedness both now

and in the future. While collaborations between water managers and agencies within countries are at least partly in place, as indicated in the survey, the difference between preparedness and proactive approaches to lower drought risk in Europe varies widely. Participants mainly in Central and Southern Europe indicated 'insufficient resources', 'lack of forcing', 'waiting for governmental advice' as a reason for not having a drought risk management plan. Across all national representatives and a majority of survey participants, there was a consensus that an EU-directive on drought risk management would be beneficial

(whether or not countries are EU-member states). Similar to the Floods Directive (EU 2007), a common strategy should only set a coarse framework, delegating specific actions to the member states and especially regulating transboundary water management during drought. An EU directive would be especially beneficial in countries where water resources management governance is not centralised (e.g. Italy and Spain), with wide procedural discrepancies among the different administrative regions and basin authorities. Recently, the Global assessment report on Drought (GAR, 2021) highlighted that adaptive risk

management and governance strategies are required as responses to complex risks such as drought, by means of actions, processes and institutions. A drought directive, following the example of the European Floods Directive, would force member states and candidate countries to act and encourage cooperation across borders addressing the regional scale of drought hazard, secure resources and funding for drought risk research and most importantly initiate a common strategy to increase drought resilience. However, not all countries fully share this view, the main reason being that a pan-European approach would not be

able to consider local specificities such as catchment physical characteristics, water infrastructure, water uses, and specific biodiversity needs. In addition, common action may be conducted at a very general, broad and political level. At operational or local level, clear and common guidelines may be needed and the challenge is to be flexible enough to cover context-specific situations.

## 5. Conclusions

The pan-European survey on drought perception and management highlighted the heterogeneity in the perception of drought hazard, impacts and management across the European continent. The reflection on the drought events in the 2018-2019 period illustrated Europe's vulnerability to drought and the variable state of preparedness to withstand drought in many countries. Even though the awareness of a future increase of drought risk is prevalent, drought is often still not considered as a risk in Central, Northern and Eastern Europe. Here, we showed that drought hazard perception matched the observed or monitored

drought hazard. In contrast, the occurrence of drought impacts does not always follow the pattern of hazard severity, and therefore requires assessment of drought beyond just the hazard. A relationship between national drought awareness and drought management strategies could not be established. Although a strong variability of drought risk management planning across the continent was evident, a common European strategy does not exist. As shown here, current national drought risk



management practices range from a fundamental lack of legislation to country wide operational drought risk management

plans. Future research might expand this survey to further explore and highlight potential benefits of a European drought directive. To foster national resilience to drought, drought management should be included in national legislation.

The key message of this study is that macro-governmental guidance by the EU is believed to be beneficial for national and international drought risk management. Such guidance should set a general framework which allows for regional flexibility of management strategies. To foster this kind of progress, sector specific databases on drought impacts, such as the EDII, are

required to show and quantify the varied impacts of past droughts and increase public awareness in order to encourage political action. Going a step further, such information should be hosted by (inter-)national drought risk monitoring systems presenting sector specific drought risk.

As the first major steps towards a more unified drought risk management in Europe, we recommend:

1-      The inclusion of a clear definition of drought in the Water Framework Directive, considering different types of

drought, as well as their spatial and temporal occurrence,

2-      The development of impact-driven, regional and sector-specific guidance on drought indices, and

3-      The formation of an inter- and transdisciplinary collaborative EU-working group focusing on drought risk management and estimation of the potential benefits and downsides of a European Drought Directive.

**Data availability**

Applied data can be accessed via http://doi.org/10.34730/ae96ed78875c4caa9ee5c25c2e2f711a.

**Competing interests**

Some authors are members of the editorial board of NHESS. The peer-review process was guided by an independent editor, and the authors have also no other competing interests to declare.

**Acknowledgments**

This research was carried out within the interdisciplinary research project DRIeR. The project is supported by the Wassernetzwerk Baden-Württemberg (Water Research Network), which is funded by the Ministerium für Wissenschaft, Forschung und Kunst Baden-Württemberg (Ministry of Science, Research and the Arts of the State Baden-Württemberg) (grant no. AZ. 7532.21/2.1.6) and Maa- ja vesitekniikan tuki ry foundation. DW acknowledges her support as part of the NERC-funded Groundwater Drought Initiative (NE/R004994/1). LJB was supported by the Natural Environment Research

Council (NE/R016429/1) as part of the UK-SCAPE programme delivering National Capability. MT, LB and JJ contributions have been supported by SustES—Adaptation strategies for sustainable ecosystem services and food security under adverse environmental conditions (CZ.02.1.01/0.0/0.0/16_019/0000797). Special thanks go to the IAHS-Panta Rhei working group "Drought in the Anthropocene" who originally designed the study.





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

**Figures and tables**

**Table 1: Drought indices and their associated drought classes. SPI, fAPAR, SM and LFI are, respectively, Standardised Precipitation Index, fraction of Accumulated Photosynthetically Active Radiation, Soil Moisture and Low Flow Index.**

| Indices | No drought | Moderate drought | Severe drought | Extreme drought |
|---|---|---|---|---|
| SPI, fAPAR, SM | > -1 | -1 to -1.5 | -1.5 to -2 | < -2 |
| LFI | 0-0.25 | 0.25-0.5 | 0.5-0.75 | 0.75-1 |




Fig. 1. Drought hazard conditions for 2018 and 2019 across the European continent according to the European Drought Observatory indicator factsheets (https://edo.jrc.ec.europa.eu/edov2/php/index.php?id=1101). Data are presented as the proportion of the country's total area under severe (or extreme) drought hazard conditions. Standardised Precipitation Index (SPI) is shown for 3, 6, 9 and 12 month accumulation periods for August (AUG), September (SEP), October (OCT) and December (DEC). SM is Soil Moisture Anomaly, LFI is Low Flow Index and fAPAR is fraction of Photosynthetically Active Radiation, all presented for maximal (MAX) proportion of severe (or extreme) hazard conditions. The timing of MAX is indicated by the number of the 10-day interval.


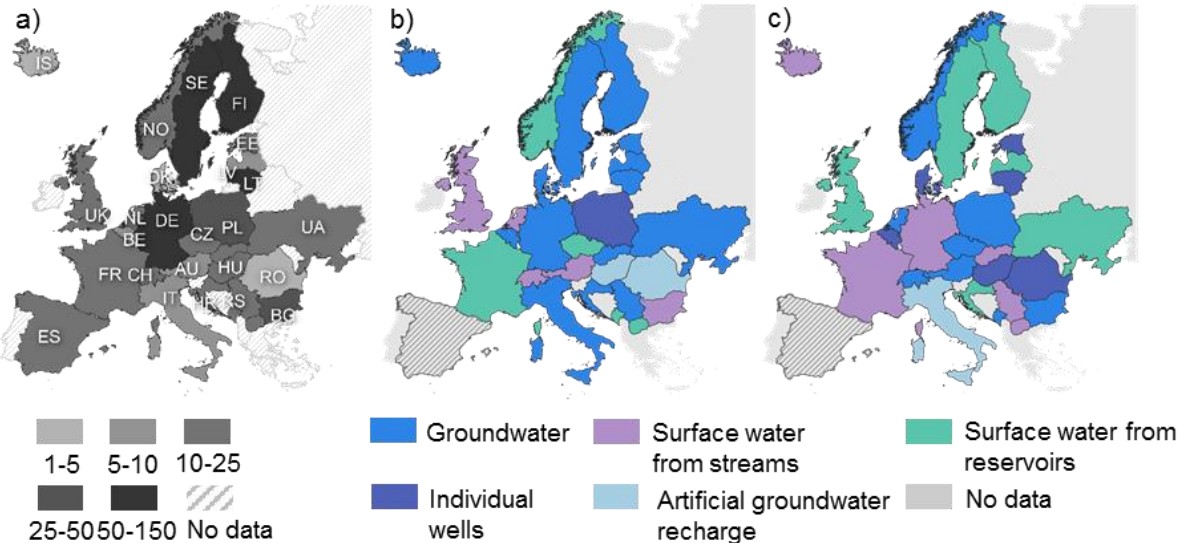

**Fig. 2. Water usage across Europe: a) number of survey participants by country, b) most important water resource by country, and**
**c) second most important water resource by country. Results are based on a pan-European survey designed by the International Association of Hydrological Science (IAHS)- Panta-Rhei "Drought in the Anthropocene" working group and conducted in 28 countries.**


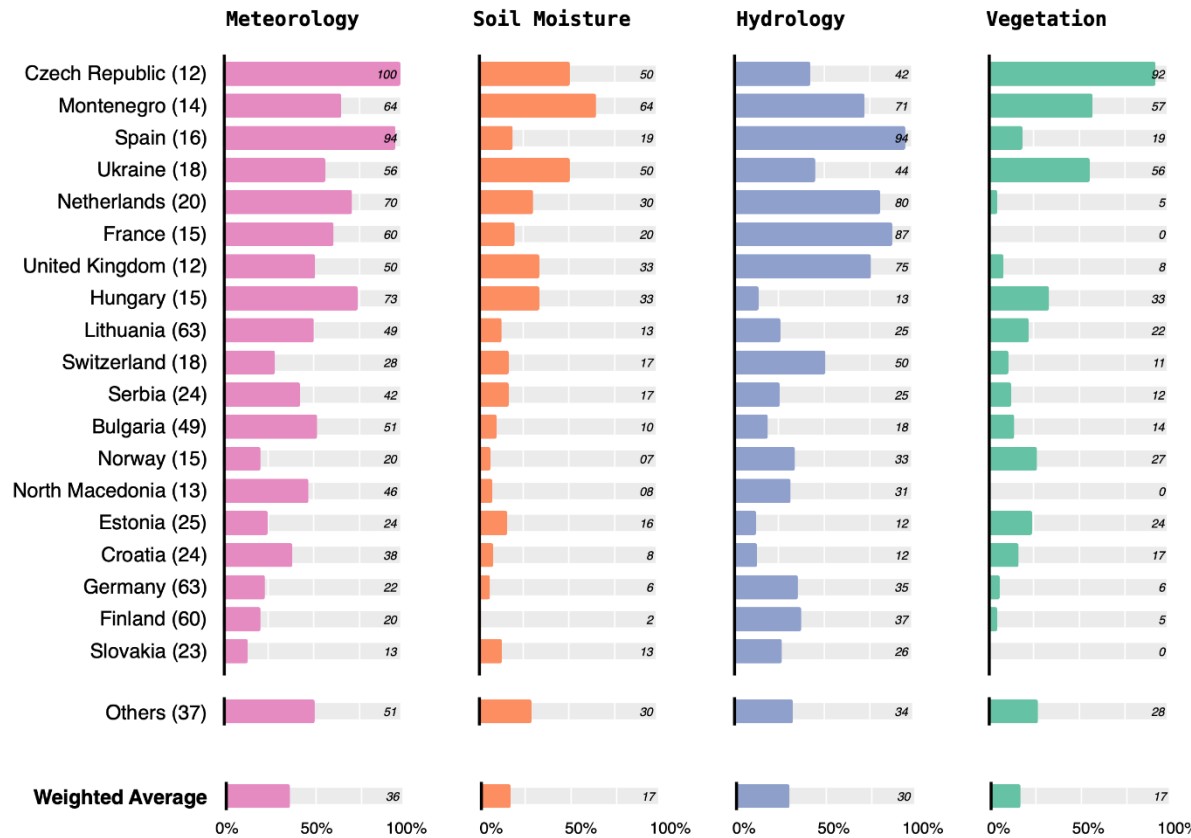

**Fig. 3. Major categories of drought indices used across Europe as a fraction of total replies per country (number of replies in**
**parentheses, total replies of 536). The mean index in each category (meteorology, soil moisture, hydrology and vegetation) is weighted**
**by the number of participants. Countries on the y-axis are sorted according to their mean index value, i.e., the highest for Czech**
**Republic, lowest for Slovakia. The category Others (n=37) comprise countries with less than 10 replies, namely Austria (9), Italy (8),**
**Belgium (6), Latvia (6), Iceland (4), Denmark (3) and Romania (1). Replies from Sweden and Poland are not considered here as**
**indices were not rated in these countries. Note that participants have different roles in their countries and thus might judge drought**
**indices differently. Results are based on a pan-European survey designed by the International Association of Hydrological Science**
**(IAHS) - Panta-Rhei "Drought in the Anthropocene" working group and conducted in 28 countries.**




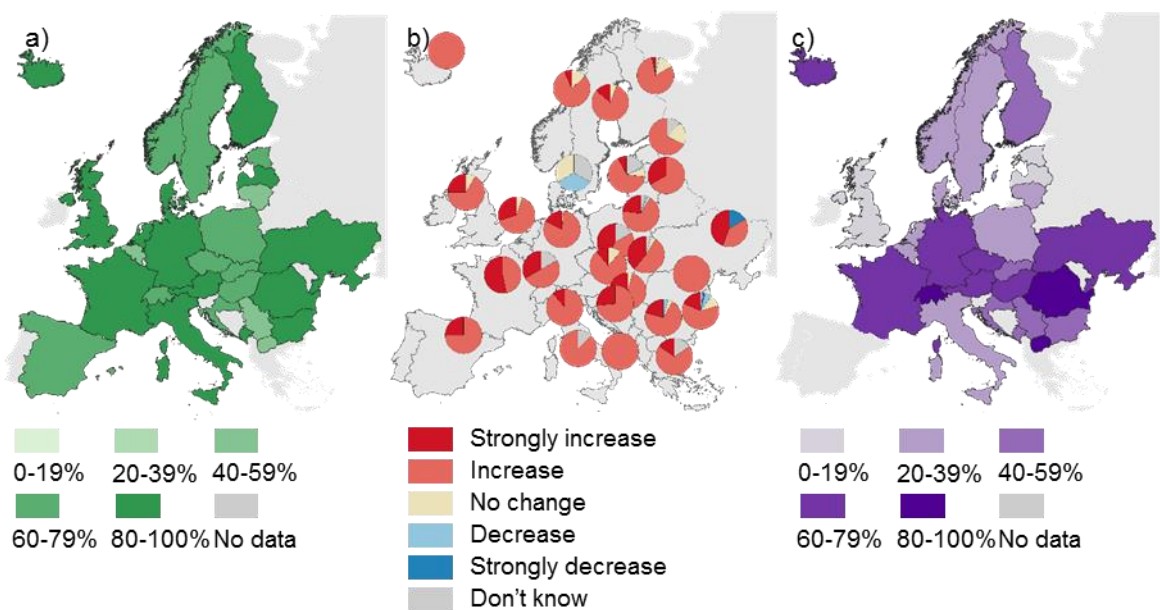

**Fig. 4. Perception of climate change effect on drought management in Europe shown as percentage of participants in pie charts responding to question: (a) whether future climate change will may affect water resources; (b) how droughts may change in future; (c) will drinking water providers in the future have to distribute water to fewer consumers due to shortages, e.g. 'rota cuts'. Results are based on a pan-European survey designed by the International Association of Hydrological Science (IAHS)- Panta-Rhei "Drought in the Anthropocene" working group and conducted in 28 countries.**


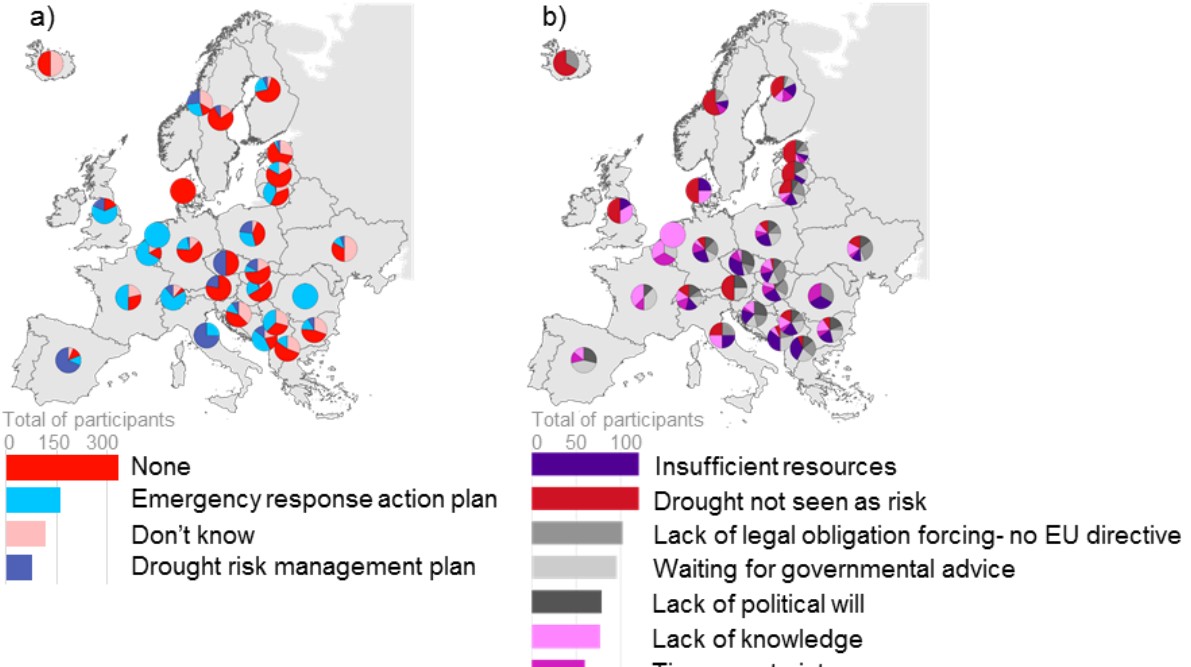

Fig. 5.

**Perception of drought risk management across Europe shown as percentage of participants in pie charts: a) distribution of drought**



**risk management plans and emergency action plans by country; b) reasons for an absence of drought risk management by country**
**and totals of selected reasons. Results are based on a pan-European survey designed by the International Association of Hydrological**
**Science (IAHS)- Panta-Rhei "Drought in the Anthropocene" working group and conducted in 28 countries.**

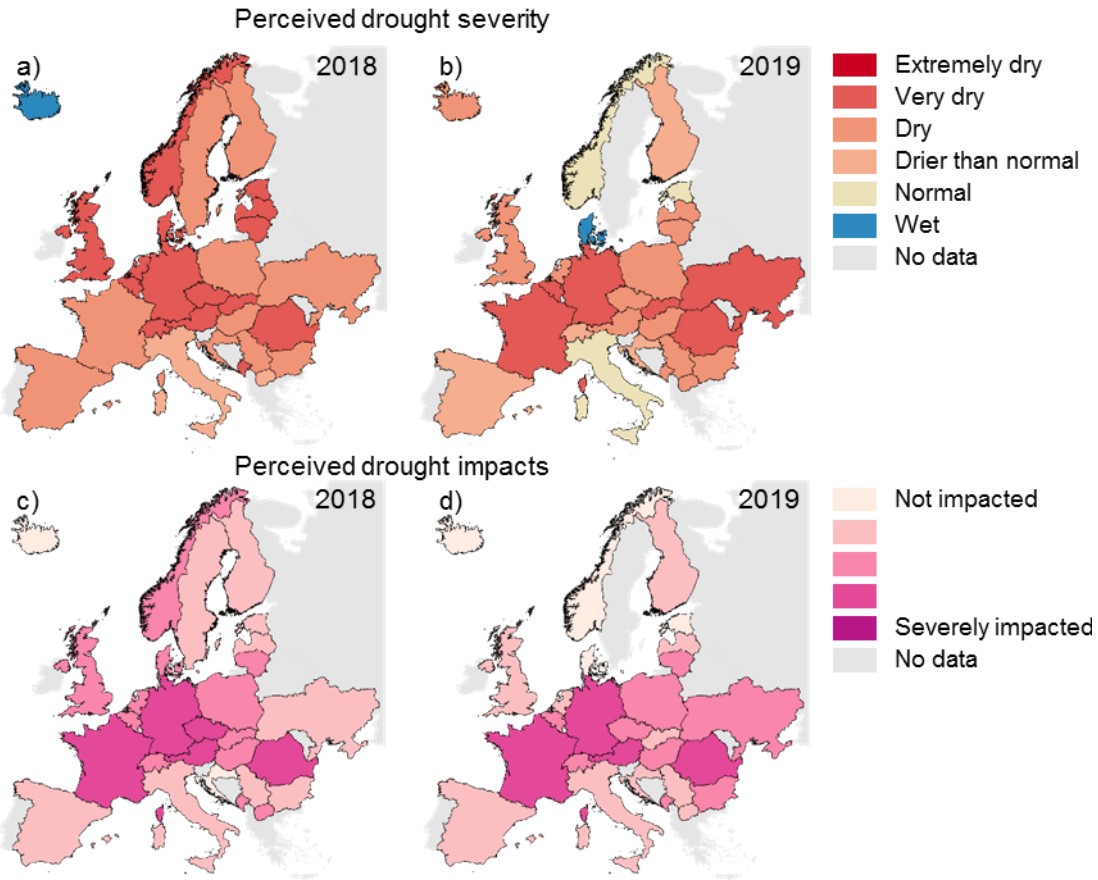

**Fig. 6. Median perception of drought severity and impacts in 2018 and 2019 across Europe. Sweden participated only in 2018. Results**
**are based on a pan-European survey designed by the International Association of Hydrological Science (IAHS) - Panta-Rhei**
**"Drought in the Anthropocene" working group and conducted in 28 countries.**




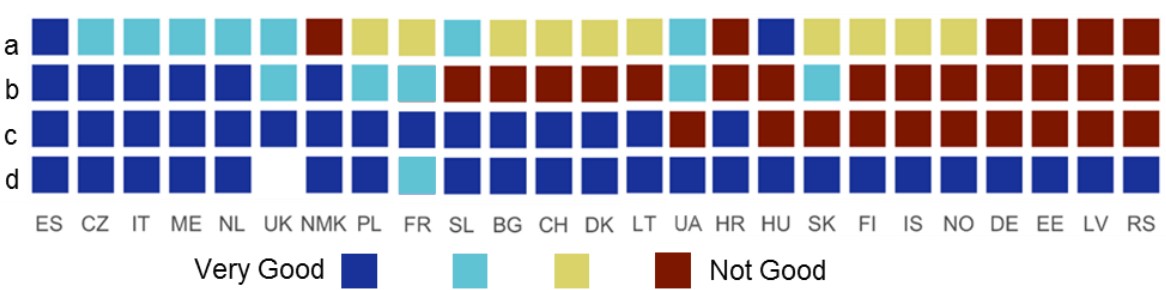

*Fig. 7.*

**695**     **National representatives' joint opinion on a) the actual state of drought management in their country, b) the existence of a country-wide drought management plan, c) the existence of national recommendations for actions in order to minimise drought risk, and d) the benefit of an EU- drought directive for their country?, ordered by score (Very good=3; Not good = 0).**