# Peer review of "Lessons from the 2018-2019 European droughts: A collective need for"

_Natural Hazards and Earth System Sciences, 2021_

## Referee Comment (RC2)

**REFEREE 2 COMMENTS**

**nhess-2021-276    Submitted on 24 Sep 2021**

**Lessons from the 2018-2019 European droughts: A collective need for unifying drought risk management**

**Veit Blauhut, Michael Stoelzle, Lauri Ahopelto, Manuela I. Brunner, Claudia Teutschbein, Doris E Wendt, Vytautas Akstinas, Sigrid J. Bakke, Lucy J. Barker, Lenka Bartošová, Agrita Briede, Carmelo Cammalleri, Lucia De Stefano, Miriam Fendeková, David C. Finger, Marijke Huysmans, Mirjana Ivanov, Jaak Jaagus, Jiří Jakubínský, Ksenija Cindrić Kalin, Svitlana Krakovska, Gregor Laaha, Monika Lakatos, Kiril Manevski, Mathias Neumann Andersen, Nina Nikolova, Marzena Osuch, Pieter van Oel, Kalina Radeva, Renata J. Romanowicz, Elena Toth, Mirek Trnka, Marko Urošev, Julia Urquijo Reguera, Eric Sauquet, Silvana Stevkova, Lena M Tallaksen, Iryna Trofimova, Michelle T. H. van Vliet, Jean-Philippe Vidal, Niko Wanders, Micha Werner, Patrick Willems, and Nenad Živković**

This is a very interesting article whose ultimate goal is to advocate the creation of a European Drought Framework Directive. For this, it is based on the analysis of the results of a survey carried out in all European countries. The supplementary material offers a series of figures that show the existence of a deep work that supports the results of the article. However, there are some aspects that are not clear enough and that I explain below.

**Section 1.3**

This section is focused on the introduction of the 2018 and 2019 droughts in Europe. Authors speak about a "summer drought" that would mean that it only lasted three months approximately. What kind of drought definition are you using here? It is not clear here that if these summer droughts refer only to a negative precipitation anomaly plus a positive temperature anomaly on summer or are they a consequence of those factors plus previous bad hydrometeorological conditions? However, in section 3 you introduce four indexes to characterize these two drought periods and it is possible to see that it is probably the same event, that started in winter 2018. In order to avoid any misunderstanding, I would suggest avoiding the expression "summer drought" here or introduce an explanation here.

**Section 2. Data**

The calculus of SPI, fAPAR, SM and LFI require data on precipitation, temperature, …. Which is the data source that have you used? And the resolution? Or are these indexes provided directly by EDO?

L. 156-164 and Table 1. The criteria used to build Table 1 and classify the category of the drought should be justified and clarified. Does exist any previous reference to justify it? To be considered into a specific category, would be necessary that the three indexes would have the same or a similar value (SPI, fAPAR, SM)? What about the different SPI indexes? Do you select the worst?

L. 167. Which criteria have you used to select the national representatives of each country? Do you consider that it constitutes a representative sample of stakeholders? What about citizens? Have do you applied any test to validate this sample from the point of view of a sociological approach?

L. 175. The sentence "National representatives of each country were selected" should be moved to the end of the previous paragraph (L.174)

**Section 3**

See comments about Figure 1 at the end of my evaluation.

L. 188. How do you use the SPI-3, SPI-6, SPI-9 and SPI-12 to define a drought? Do you apply each one to each month?

L.220-L.225. I consider that it is not necessary to explicit the name of all the countries affected by the drought when it is possible to synthesize it with the only reference to the Balcanic Peninsula. It is to say, the text "In south- eastern Europe, winter 2018/2019 precipitation deficits were detected across much of the Balkan Peninsula, i.e., Croatia, Albania, Slovenia, North Macedonia, Montenegro and Hungary, as well as in Slovakia. Serbia was already affected by the summer drought in 2018, which persisted throughout the winter. In Ukraine, Moldova and Romania, the 2018 event was moderate in the second half of the year and further rainfall deficits accumulated during winter, which led to rising soil moisture deficits from summer 2019 to the end of 2020", could be replaced by "In south-eastern Europe, winter 2018/2019 precipitation deficits were detected across much of the Balkan Peninsula. In Ukraine, Moldova and Romania, the 2018 event was moderate in the second half of the year and further rainfall deficits accumulated during winter, which led to rising soil moisture deficits from summer 2019 to the end of 2020".

L. 227. See my comments about Figure 2.

**Discussion**

L. 342-354. Why do you introduce here a paragraph about some meteorological/climatic conditions related with some droughts in some parts of Europe? There is not any connection with the previous sections, neither with the subject of the paper. I would propose to delete them because it only offers a very partial image.

L. 359. It would be useful to build a figure that could corroborate the sentence "In general, the hazard severity perceived by the surveyed stakeholders corresponded well with the hazard severity monitored by the EDO". It is not easy to compare Figure 1 with Figure 6. You can create a graph x/y, Perceived/monitored.

L. 387. Add a parenthesis before Fig.

L. 390. Delete the initials of the name in the reference "Hervás-Gámez, C., & Delgado-Ramos, F.,"

**Figures**

The quality of figures is not good. The main problem lays in the labels.

Figure 1. Years 2018 and 2019 placed at the left side can be deleted. It is enough to write it in the legend of the figure. Delete the text into the figure ("Proportion of country…."). This figure

has been built by you or is from EDO? What about "The timing of MAX is indicated by the number of the 10-day interval"? Any reference to MAX neither to 10-day interval is included in the main text. Why have you selected those months and SPI accumulated values? (i.e. October, SPI-9).

Figure 2. It is not needed to write the capital letters to identify each country. As far as I understand the information about the main water usage is provided by the people interviewed? The results really caught my attention. Would it be possible to check if that perception is correct? Why there are no data for Spain? Following the text there is enough information.

Figure 4. Although the legend says "Perception of climate change effect on drought management in Europe shown as percentage of participants in pie charts responding to question", only Figure 4b shows pie charts.

---

## Author Response (AR1)

Dear reviewers and respondents from the public discussion,

Many thanks for time and efforts taken to review and discuss our manuscript. The most fiery debated point of our manuscript was, unfortunately, more of a political nature rather than scientific. Thus, we want to address this first.

We, the authors, fully accept the sovereignty of Kosovo. There is no discussion on this point. Our choice was not politically motivated. Our study is based on the NUTS-level classification system (and GIS data) as provided by Eurostat for the year 2016 as our study was initiated in 2019. The NUTS-classification system and GIS data did not include Kosovo, which is due to issues of the European Commission and Eurostat. Accordingly, we structured our pan-European survey on the NUTS regions and countries as given. Even though we would like to adjust our maps to the raised issue, we just simply can't. The questionnaires were not suited to this split, and also cannot be repeated.

In consultation with NHESS, considering the Copernicus Publications requirement on behalf of the EGU journals, we adapted the manuscript. To explain "the issue" we added the following to the main text data chapter "Kosovo has not investigated disaggregated from Serbia (please see disclaimer)" and a more explicit explanation to the disclaimer"We the authors fully accept the sovereignty of Kosovo. This study is based on the NUTS-level classification system (and GIS data) as provided by Eurostat for the year 2016. (The study was initiated in 2019). The NUTS-classification system and GIS data did not include Kosovo. Accordingly, the pan-European survey followed the NUTS regions and countries as given." Furthermore, we added the borders of Kosovo as dashed lines to the all maps.

**Dear Reviewer #1,**

Many thanks for your thoughtful review of our manuscript. We highly appreciate your efforts and your positive evaluation of our work.

We extracted the more general question from the first part of the review and will reply to these first, followed by the reply to the "specific comments".

**General comments**

*There is a very large heterogeneity in how countries declare drought situations. Also, the expectation on the impacts of climate change, has no relationship to how this is considered in policies. However, there is some link to climate, respondents from non-European countries see less need to increase regulation to prepare for climate change.*

Dear Reviewer #1, many thanks for this general comment. Since our survey was on European countries only we assume that you were pointing on non-EU (European Union) members. A clear tendency towards this is not in our date nor seen during data analysis. The expected need to regulate water distribution more with respect to climate change might rather be influenced by the countries exposition to drought in the future and drought management strategies actually implemented.

*The results of the survey are very heterogeneous, due to different factors. First, that the pedo-climate and socio-economic conditions of different countries are very different. Second, droughts are never pan-European, they always affect different regions differently, in this case, the Mediterranean was much less affected than Central, Northern and Eastern Europe. I wonder also about the representativity of the survey, does it represent well all relevant sectors in all countries?*

Dear reviewer, many thanks for this comment. Indeed, pedo climate and socio-economic factors different over the continent. These and the fact that droughts are never (as far as we know) pan-European, could be approved by our findings. Our hazard analysis showed that for 2018 and 2019

drought events, the centre of the drought was in central-northern Europe, shifting towards eastern Europe in 2019. Furthermore, our portrayal of drought impacts perceived by sector indicated national differences, which might be due to the effects of the hazard and different underlying vulnerabilities.

As we indicated in the data section – line 176:" The survey targeted representatives of water management organisations and water agencies. Survey respondents were selected by the national representatives aiming to provide a balanced view of national opinions and drought management practices (or actions), as well as local and regional knowledge within each country" Furthermore we picked up on the representativity issue in the discussion line 389 "Our pan-European survey reflects the opinions of water professionals belonging to mostly the public sector and publicly owned companies. The perspectives of other citizens, local stakeholders, private companies and NGOs were less well represented. Nevertheless, the fraction of respondents'-affiliation differs among countries and can thus have had an influence of the herein generalised portrayal of drought risk. A statistical relation between affiliation and "other" replies (such as drought management, or reason for a lack of DRM) could not be found. Furthermore, sectoral and regional perceptions of drought risk might differ." We hope that this answers your question.

*In general, the hazard severity perceived by the surveyed stakeholders corresponded well with the hazard severity monitored by the EDO, which is good news. The discrepancies can be due to sectorial differences, scale differences between impact indicator and affected sector and also to the pertinence of a standardized index to represent potential impacts.*

Thank you for this comment. Indeed, we were also glad to see a coherence with the EDO monitoring.. We will list potential causes for the differences still seen in a generic way. We also explored potential causes in the discussion and gratefully added your thoughts to it: "These discrepancies could be attributed to the low awareness of the stakeholders for the drought conditions at a larger scale or the impact across different sectors, or the discrepancy between impact indicator and affected sector. Alternatively, standardised drought indices not effectively representing drought conditions everywhere may also contribute to these discrepancies. In higher latitude countries, a strong negative rainfall anomaly does not necessarily imply a deficit in water availability for e.g. plant water uptake or public water supply as storages are usually replenished after the snow melt period. (Cammalleri et al. 2016). As such, meteorological drought indices may not be appropriate to predict impacts and consequences for management of hydrological or agricultural (soil moisture) droughts."

*Concerning standardized indices, the article has a very good discussion on the topic, which states that "drought indices may not be appropriate to predict impacts and consequences for management of hydrological or agricultural (soil moisture) droughts". Later it says "an effective implementation of macro-regional drought risk management requires a more holistic interdisciplinary view".*

Many thanks for this suggestion. In line 384 we added "Thus, drought cannot be declared by a single index only, the entire water cycle has to be considered, since droughts in different parts of the water cycle can lead to different impacts. Such a holistic view should start with initial meteorological drought (e.g. lower than normal precipitation often combined with higher than normal evaporation), causing a deficit in soil moisture, and if sustained for a sufficient time, may manifest itself as a hydrological drought (i.e. a deficit in streamflow and groundwater).

*The situation of each country, in terms of drought management, depends on the impacts and frequency of drought in this country. There is a strong memory effect too, so recent droughts incentivize the development of measures.*

Many thanks for your comment. Indeed, memory effects are postulated by various authors and are supported by our findings. Nevertheless, we also suppose that engagement can increase awareness, which can partly overcome the memory effect.

*In page 13 there is a discussion on the effects and needs of an EU-level drought directive. It is unclear if this discussion is derived from the survey, or it is the opinion of the authors. This should be clarified.*

Many thanks for your comment. We now clarified this in line 418: "The majority of the survey participants and all national representatives agreed that a pan-European drought management approach would support national and cross-boundary drought preparedness both now and in the future."

**Specific comments:**

*Maybe the data on the most important water resources by country can be organized in more homogeneous and clear categories:*

• *Figures 2b and 2c show that there is no data on the most important water resource in Spain. This seems strange to me, as there are between 10 and 25 respondents in this country. Then, I have seen in the text that "this question was adapted to national specificities and resulted in #1 regulated surface water and #2 groundwater". It seems to me that "regulated surface water" is mainly "Surface Water from Dams", so it could go into this category. I think this can be improved for greater clarity.*

• *In two countries, artificial groundwater recharge (Figure 2b) is the main source of water, but the artificially recharged water comes from somewhere. Does it come from the river? Does it come from water treatment plants? My question is, is artificial groundwater recharge a water resource, or it is a method to distribute the resource? Also, I don't understand why individual wells are not included in the same category as groundwater.*

*So, it seems to me that:*

*1. The number of categories can be reduced (groundwater and individual wells).*

*2. The Spanish categories can be assigned to the same categories as the rest of Europe.*

*3. Clarify what "Artificial recharge" means in this context, because the water injected must come from some other resource.*

Many thanks for your specific comments.

1. The types of water usage were defined by the national representatives at the initial phase of this survey. Even though the intention was to cover the major water usages in Europe, each participant was also free to add additional sources. Summarising the variety of uses was difficult. Since the respondents were asked to specifically reply to the listed categories, we refrain from summarising these. To clarify this we added "The participants were asked to rank a selection of water resources, but were also free to add additional ones." to chapter 3.2

Of course, defining the correct source is not an easy task. Especially with regard to groundwater and individual wells the difference was groundwater usage by public water supply (large scale) and water supply by individual wells. Hence, a strong message comes with separating between the two.

2. Indeed, having no data for Spain, which seems strange at a first view. Unfortunately, the questionnaire (which was translated to the national language by the national representatives) was "adapted" to the national needs and it was formulated differently. Even though we initially decided to

have a uniform survey for all countries, this happened. Since we noticed this after the finalisation of all questionnaires, there was no chance to redo it. Reconsidering the Spanish case "regulated surface water" is equivalent to the category "surface water from reservoirs". Accordingly we have followed the Reviewer's suggestion and have used the existing categories for the rest of Europe to represent Spain's data. Please see the changes in Figure 2, as well as in the text chapter 3.2: "In the case of Spain, the questionnaire was adapted to national specificity and resulted in less water-usage categories; here "regulated surface water" falls in the category "surface water from reservoirs". Accordingly, water resources ranks are #1 regulated surface water and #2 groundwater."

3. The terminology of "artificial recharge". As you correctly defined, the water must come from somewhere, such as rivers, water transfers or treatment plants. In the survey, we did not ask to specify the source of water for the artificial recharge and thus, we cannot specify this term further.

To clarify these point we added "The participants were asked to rank a selection of water resources, but were also free to add additional ones. The sources of "artificial recharge" were not specified" to chapter 3.2.

*Concerning the indices by country, the text says (Page 8, 243): "About 40% of all participants did not have an operational drought definition in their municipality or company, and a further 15% did not know whether there was one". I'm surprised by the words "municipality" and "company". What kind of managers were these? Are these workers from municipal water supply? What about water managers at the regional, basin and national levels? They don't work for municipalities, nor companies, they work for governmental institutions.*

Many thanks for raising this question. As we stated in section two- data "National representatives of each country were selected and assigned responsibility to translate, distribute and evaluate the survey and all associated communication and feedback. The survey targeted representatives of water management organisations and water agencies. Survey respondents were selected by the national representatives aiming to provide a balanced view of national opinions and drought management practices (or actions), as well as local and regional knowledge within each country." Accordingly, a general differentiation between public and private water suppliers/ managers is difficult, since water management is organised very differently, even within countries. E.g. for some cases in Germany, "public companies" might be owned by the government or water is managed at different governmental levels. In our survey, we also asked for the location/spatial scale of the respondents work, as well as their position in and the role of the company. Unfortunately, a generalisation of the over 700 replies was not possible. To provide an impression of the respondents' affiliation, a table showing number of participants by affiliation and country is presented in Tab.S2 and referenced in the text (Line 255). To clarify the issue of "municipality or company", we changed this to "public and private organisations".

*The results are very heterogeneous, due to a lot of reasons. I wonder if the results are also different between countries due to biases in the different kinds of participants in each country. Maybe in one country the respondents are more biased towards water managers, in others toward water supply utilities, etc. I would like to see a table or chart that shows the kinds of participants per country.*

Dear reviewer #1, many thanks for your comment which follows up on the previous one. Indeed, it is likely that the national perspective is dominated by respondent's affiliation, position and duties. For more detailed insights, we refer to Table S2. A statistical relation between affiliation and "other" replies (such as drought management, or reason for a lack of DRM) could not be found. These results are not shown here. To clarify this caveat we revised this part of the text (line 389) as follows: "Our pan-European survey reflects the opinions of water professionals belonging to mostly the public sector and publicly owned companies. The perspectives of other citizens, local stakeholders, private

companies and NGOs were less well represented. Nevertheless, the fraction of respondents'-affiliation differs among countries and can thus have had an influence of the herein generalised portrayal of drought risk. A statistical relation between affiliation and "other" replies (such as drought management, or reason for a lack of DRM) could not be found. Furthermore…"

*The discussion in the last paragraph of page 11 and first paragraph of page 12 is very good. Earlier in the article it is said that, in Spain, one of the countries which has a more developed drought policy, each basin has defined its own drought and water scarcity indices, which are defined according to local needs. I would like to know if this is the kind of holistic approach the authors have in mind or if they are thinking on a different approach. I think this is the conundrum to solve if we want to have an EU-scale drought policy that makes sense and which the users trust. Maybe the authors could comment on this. But I would understand if they want to be more diplomatic and want to avoid comparing the policies of different countries.*

Dear reviewer#1, as already introduced, you are here hitting a critical point that was thoroughly discussed in our consortium. What we could conclude is that a "general strategy" has to exist at a superior governmental level (for Europe, this should be the EU). Specific plans, actions, probably even indicators, will have to be adapted to the regional and maybe local needs. An idea that is similar to the Flood Risk Framework. Spain is a good European example that has already implementing such practice for drought: general boundaries for management and "freedom" of actions for the sub-national level. At the moment, it is key to begin aligning approaches and implementing such a strategy! In the long term, a supra national/ pan-European strategy from early on would ease things. But if this is not possible (yet?), there is no reason a country, municipality or company cannot implement drought management and risk plans at a lower decision unit.

To underline our statement we added "Accordingly, a pan- European approach should also be tailor-made to be trusted by the users. Therefore specific indicators and actions can be tailored to local situations and needs, but a general framework should be guiding the application of these." to the end of the discussion (line 436).

*It must be made more clear if the discussion about an EU drought directive found in page 13 reflects the opinions of the survey respondents or the author's. For example, and this is just an example, it is said "However, not all countries fully share this view", based on what? How did the authors arrive to that conclusions. So, I'm not sure if this rather interesting discussion belongs to this paper or to an opinion piece. Author's should edit this section to make more clear on what basis they affirm what they affirm and how the survey shaped these statements.*

Many thanks for this comment. Indeed, this was not clear enough. "Countries" is changed to "respondents of the survey". (Line 433)

*Technical corrections*

*Figure 2.a would be easier to read if it was in color, instead of grayscale.*

Many thanks for pointing this out. We had figure 2a coloured in previous internal drafts, but the number of colours used throughout the manuscript became too high and so we decided to use grayscale for this figure. It is thus an easy to assess gradial scale now.

**Dear Reviewer #2,**

Many thanks for your thoughtful review of our manuscript. We highly appreciate your efforts and your positive evaluation of our work, we respond to your comments below.

*Section 1.3: This section is focused on the introduction of the 2018 and 2019 droughts in Europe. Authors speak about a "summer drought" that would mean that it only lasted three months approximately. What kind of drought definition are you using here? It is not clear here that if these summer droughts refer only to a negative precipitation anomaly plus a positive temperature anomaly on summer or are they a consequence of those factors plus previous bad hydrometeorological conditions? However, in section 3 you introduce four indexes to characterize these two drought periods and it is possible to see that it is probably the same event that started in winter 2018. In order to avoid any misunderstanding, I would suggest avoiding the expression "summer drought" here or introduce an explanation here.*

Many thanks for pointing this out.

Droughts are defined by index specific thresholds as presented in Table 1, starting from "moderate drought". Indeed, "summer drought "is misleading and was intended to convey that some of the most severe deficits were seen over the summer months. But as the drought conditions extended beyond the summer we have removed "summer".

*Section 2. Data: The calculus of SPI, fAPAR, SM and LFI require data on precipitation, temperature, …. Which is the data source that have you used? And the resolution? Or are these indexes provided directly by EDO?*

Many thanks for this comment. We have clarified this point in the revised paper and have changed the data description to "Detailed information on the drought indices applied herein can be found in the EDO indicator factsheets (https://edo.jrc.ec.europa.eu/edov2/php/index.php?id=1101)."

*L. 156-164 and Table 1. The criteria used to build Table 1 and classify the category of the drought should be justified and clarified. Does exist any previous reference to justify it? To be considered into a specific category, would be necessary that the three indexes would have the same or a similar value (SPI, fAPAR, SM)? What about the different SPI indexes? Do you select the worst?*

Many thanks for raising this point. Indeed, the associated drought classes could have been explored in more detail. As you know, these thresholds are often an "arbitrary choice" and focus only on the hazard anomaly. As SPI, fAPAR and SM are standardised indices, their values come with a certain return period of the drought conditions. As such, the classification for these three standardised products: SPI, fAPAR, SM is taken from McKee et al. (1993). "The Low-Flow Index (LFI) is computed from the daily streamflow values produced by the LISFLOOD hydrological model". The drought classification scheme used for LFIs taken from the European Drought Observatory and is explained in more detail in the relevant fact sheet (https://data.jrc.ec.europa.eu/dataset/fd18fd1d-1af4-443a-b624-e19001b91f49).

To clarify this, we added "To increase comparability of the four indices, the EDO data was further classified into categorical drought classes: no drought, moderate drought, severe drought and extreme drought. The standardised products SPI, fAPAR and SM are categorised following McKee et al. (1993) (Table 1), the Low-Flow Index (LFI) is computed from the daily streamflow values produced by the LISFLOOD hydrological model". The drought classification scheme used for LFI is taken from the European Drought Observatory ( Table 1). These drought classes are in operational use at the EDO. Furthermore, the fAPAR was restricted to the warm season in Europe from April to August and

was not monitored for Iceland. Detailed information on the drought indices and drought classes applied herein can be found in the corresponding EDO indicator factsheets (https://edo.jrc.ec.europa.eu/edov2/php/index.php?id=1101).” to the text (line 161).

*L. 167. Which criteria have you used to select the national representatives of each country? Do you consider that it constitutes a representative sample of stakeholders? What about citizens? Have do you applied any test to validate this sample from the point of view of a sociological approach?*

Dear Reviewer #2, many thanks for raising these questions. Finding contributors was not an easy task to do. The initial idea was to have representatives affiliated to science or governmental agencies. Doing so, we expected a neutral point of view and comprehensive knowledge on the different aspects we were interested in. Furthermore, we expected such persons to be well networked and thus constitute a representative sample of stakeholders. The network of national representatives developed out of our active Panta Rhei- Drought in the Anthropocene group but also partly from the Euro-FRIEND low flow group. For countries with no contacts, we screened national and international publications in the wide field of drought research. Clearly visible in Figure 2a, we did not manage to find support in each European country. The Covid situation of the past years complicated the contribution from many national representatives.

To clarify the selection procedure, we added the following text to chapter 2:

“National representatives of each country were selected and assigned responsibility to translate, distribute and evaluate the survey and all associated communication and feedback. The network of national representatives developed out of our active Panta Rhei- Drought in the Anthropocene group but also partly from the Euro-FRIEND low flow group. The idea was to have representatives affiliated to science or governmental agencies. Doing so, we expected a neutral point of view and comprehensive knowledge on the different aspects we were interested in. Furthermore, we expected such persons to be well networked and thus constitute a representative sample of stakeholders.” (line 171)

We did not test this sample from a sociological perspective. The number of replies are strongly biased by country. Likely due to very different reasons which we can only speculate about. Nevertheless, we realised that our method was not optimal, but still considered a representative and well informed sample of national representatives

*L. 175. The sentence “National representatives of each country were selected” should be moved to the end of the previous paragraph (L.174)*

Please see comments and changes to the previous comment. Thank you.

*L. 188. How do you use the SPI-3, SPI-6, SPI-9 and SPI-12 to define a drought? Do you apply each one to each month?*

Dear reviewer #2, many thanks for raising this question. Following up on a previous comment, droughts are defined as introduced in Table 1. The time steps of SPI are monthly, the other indices fAPAR, SM and LFI are updated every 10 days. Accordingly, the definition of drought follows this frequency.

*L.220-L.225. I consider that it is not necessary to explicit the name of all the countries affected by the drought when it is possible to synthesize it with the only reference to the Balcanic Peninsula. It is to say, the text "In south- eastern Europe, winter 2018/2019 precipitation deficits were detected across much of the Balkan Peninsula, i.e., Croatia, Albania, Slovenia, North Macedonia, Montenegro and Hungary, as well as in Slovakia. Serbia was already affected by the summer drought in 2018, which persisted throughout the winter. In Ukraine, Moldova and Romania, the 2018 event was moderate in the second half of the year and further rainfall deficits accumulated during winter, which led to rising soil moisture deficits from summer 2019 to the end of 2020", could be replaced by "In south-eastern Europe, winter 2018/2019 precipitation deficits were detected across much of the Balkan Peninsula. In Ukraine, Moldova and Romania, the 2018 event was moderate in the second half of the year and further rainfall deficits accumulated during winter, which led to rising soil moisture deficits from summer 2019 to the end of 2020".*

Dear Reviewer #2, many thanks for this suggestion. We adapted the manuscript accordingly.

L. 227. See my comments about Figure 2.

*Please see our answer regarding Fig. 2 below.*

*Discussion*

*L. 342-354. Why do you introduce here a paragraph about some meteorological/climatic conditions related with some droughts in some parts of Europe? There is not any connection with the previous sections, neither with the subject of the paper. I would propose to delete them because it only offers a very partial image.*

Thank you for your suggestion. However, we find it important to provide an explanation for drought spatial heterogeneity from a meteorological and a climatological perspective. Additionally, as the climate studies revealed an increase in the frequency of drought related to circulation patterns, we found it appropriate to mention this in the section of discussion.

*L. 359. It would be useful to build a figure that could corroborate the sentence "In general, the hazard severity perceived by the surveyed stakeholders corresponded well with the hazard severity monitored by the EDO". It is not easy to compare Figure 1 with Figure 6. You can create a graph x/y, Perceived/monitored.*

Thank you for this suggestion. We revised Fig. 1 and presented there now lines or coloured areas for the different drought indices in the two drought years 2018 and 2019. We hope with that a comparison across drought indices and drought perception is now much easier. A direct comparison of perceived and monitored hazard conditions seems impractical due to the different temporal resolution of data.

*L. 387. Add a parenthesis before Fig.*

Dear Reviewer #2, many thanks for this suggestion. We adapted the manuscript accordingly.

*L. 390. Delete the initials of the name in the reference "Hervás-Gámez, C., & Delgado-Ramos, F.,"*

Dear Reviewer #2, many thanks for this suggestion. We adapted the manuscript accordingly.

*Figures*

*The quality of the figures is not good. The main problem lays in the labels. Figure 1. Years 2018 and 2019 placed at the left side can be deleted. It is enough to write it in the legend of the figure. Delete the text into the figure ("Proportion of country…."). This figure has been built by you or is from EDO? What about "The timing of MAX is indicated by the number of the 10-day interval"? Any reference to MAX neither to 10-day interval is included in the main text. Why have you selected those months and SPI accumulated values? (I.e. October, SPI-9).*

Thank you for this important comment. Yes, the added numbers on the countries are too much information. Therefore, we reconceptualised the figure showing now the values of the three indices over the years 2018 and 2019.

*Figure 2. It is not needed to write the capital letters to identify each country. As far as I understand the information about the main water usage is provided by the people interviewed? The results really caught my attention. Would it be possible to check if that perception is correct? Why there are no data for Spain? Following the text there is enough information.*

Dear Reviewer #2, many thanks for your important comment. This point was also raised by reviewer #1. Data for Spain exists. Unfortunately, the Spanish- questionnaire (which was translated to the national language by the national representatives) was "adapted" to the national needs and it was formulated differently. Even though we initially decided to have a uniform survey for all countries, this happened. Since we noticed this after the finalisation of all questionnaires, there was no chance to redo it. Reconsidering the Spanish case "regulated surface water" is equivalent to the category "surface water from reservoirs". Accordingly we have followed the Reviewer's suggestion and have used the existing categories for the rest of Europe to represent Spain's data. Please see the changes in Figure 2, as well as in the text chapter 3.2:"In the case of Spain, the questionnaire was adapted to national specificity and resulted in less water-usage categories; here "regulated surface water" falls in the category "surface water from reservoirs". Accordingly, water resources ranks are #1 regulated surface water and #2 groundwater." Furthermore, we removed the capital letters in Figure 2.

*Figure 4. Although the legend says "Perception of climate change effect on drought management in Europe shown as percentage of participants in pie charts responding to question", only Figure 4b shows pie charts.*

Dear Reviewer #2, many thanks for this suggestion. We adapted the manuscript accordingly.

---

## Author Response (AR2)

Dear Editorial-team,

Many thanks for your positive evaluation and your excellent assistance and guidance through the submission process.

As you requested, we applied the minor changes in Figure1.

Furthermore, as I already informed you, additional changes were applied, as requested by co-authors:

- The dashed line of Kosovo was replaced by a different kind of dashed line. Now, the dashes are more visible.
- The disclaimer was shortened to the most diplomatic consensus. Accordingly no country (or not country) is mentioned. "This study is based on the NUTS-level classification system (and GIS data) as provided by Eurostat for the year 2016. Accordingly, the pan-European survey followed the NUTS regions and countries as given."